# Rethinking Contrastive Language-Image Pre-Training for Medical Cross-Modal Retrieval: Beyond One-to-One Correspondence

## Abstract

Cross-modal retrieval using contrastive language-image pre-training (CLIP) has achieved remarkable success, and also in medical applications. While effective, current CLIP-based approaches for medical image-report retrieval overlook critical differences between medical and natural image-text pairs. Unlike concise natural image captions, medical reports are long, multi-faceted descriptions of their paired images. Furthermore, similar pathological patterns frequently recur across different medical cases. These characteristics challenge CLIP's image-text alignment paradigm, which struggles with lengthy reports and ignores inter-case similarities. To address these limitations, we propose two innovations: HIP-InfoNCE, a contrastive loss that aligns holistic images with multiple stochastic masked views of their corresponding reports, and text-aware label smoothing, which incorporates inter-report semantic similarity into supervision, which incorporates inter-report semantic similarity into supervision. Extensive experiments demonstrate that our approach outperforms existing methods by significant margins and achieves state-of-the-art performance.

## 1 Introduction

Cross-modal retrieval of medical images and reports is increasingly important in clinical practice, medical education, and research. This task enables efficient case search, facilitates second opinions, and supports evidence-based decision-making. With the rapid growth of medical imaging, the ability to retrieve relevant cases from large databases such as MIMIC-CXR (Johnson et al., 2019) is becoming indispensable. MIMIC-CXR alone contains over 377,000 chest X-rays paired with 227,000 radiology reports, underscoring both the scale and complexity of the problem.

Recent advances in contrastive language-image pre-training (CLIP) (Radford et al., 2021) have achieved remarkable success in natural image-text retrieval, inspiring numerous medical adaptations (Liu et al., 2023a; Huang et al., 2024a; Chen et al., 2024; Liu et al., 2025). However, directly transferring CLIP to the medical domain overlooks key domain-specific properties of image-report pairs. As illustrated in Figure 1, our analysis identifies two fundamental challenges: ❶ **Holistic image versus multi-faceted report.** Unlike concise captions in CLIP's training on natural images (typically 10-15 words), medical reports are comprehensive documents describing multiple aspects of their linked images. MIMIC-CXR radiology reports average 30-40 words in findings sections alone, with complete reports often exceeding 70-80 words. These reports systematically address anatomical structures, pathological findings, clinical observations, and comparative assessments, forming a rich yet complex relationship between image and text modalities. For example, a single chest X-ray may be described through cardiac silhouette assessment, pulmonary parenchyma evaluation, pleural space analysis, and skeletal structure observations. This one-to-many semantic alignment fundamentally differs from CLIP's atomic image-text pairing paradigm, causing its coarse alignment strategy to miss critical granular relationships. ❷ **Overlapping semantics across cases.** Medical images and reports also exhibit frequent partial similarity. For instance, in MIMIC-CXR, over 43.5% of reports contain the phrase "no pleural effusion", and many mention recurring observations like "enlarged heart" or "atelectasis" (Tanno et al., 2025; Zhu et al., 2025). Thus, different image-report pairs may share non-trivial semantic overlap. Yet CLIP assumes a strict one-to-one correspondence between images and texts through binary supervision with an identity matrix, ignoring nuanced inter-case similarities and constraining representation learning.

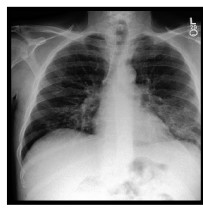
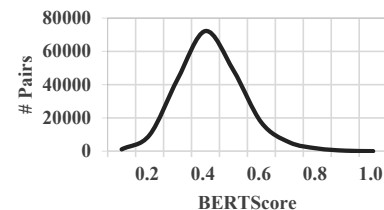

**Report:**

*Heart size is normal. The mediastinal and hilar contours are unremarkable. Ill-defined patchy opacities are noted in both lung bases, left more so than on the right, concerning for infection or aspiration. No pleural effusion or pneumothorax is visualized. The pulmonary vascularity is not engorged. No acute osseous abnormality is identified. Patchy ill-defined opacities in both lung bases, left more so than right concerning for infection or aspiration.*

Figure 1: Characteristics of medical image-report pairs. Left: Reports are semantically rich and multi-faceted; each text view aligns with the image. Right: Cases often overlap semantically. Pairwise BERTScore on 200k randomly sampled MIMIC-CXR reports confirms this.

To address these challenges, we propose a novel framework for medical image-report pairs with two methodological innovations: ❶ **HIP-InfoNCE (holistic image, partial-text InfoNCE).** We introduce a contrastive loss that aligns an image with multiple masked text views of its paired report. Rather than explicit sentence-level decomposition (Xu et al., 2024; Barrow et al., 2020), we employ stochastic masking to generate diverse, semantically partial views without requiring report parsing or domain expertise. This strategy avoids brittle text segmentation, encourages robustness, and yields stronger empirical performance. ❷ **Text-aware label smoothing.** We replace CLIP's rigid identity matrix with soft labels reflecting varying degrees of semantic similarity across cases. This smoothing is dynamically informed in a batch-wise manner, enabling the model to learn more nuanced image-text relationships.

Our main contributions are:

- We analyze domain-specific challenges of medical cross-modal retrieval, explaining why direct use of CLIP's architecture is insufficient.

- We propose HIP-InfoNCE, a loss that models the holistic-to-partial nature of image-report alignment through stochastic masked text views.

- We introduce text-aware label smoothing, which incorporates inter-report similarities for improved contrastive learning.

- Our approach achieves state-of-the-art performance, outperforming existing methods across multiple evaluation metrics.

## 2 METHOD

### 2.1 PRELIMINARY

The CLIP framework (Radford et al., 2021) consists of an image encoder and a text encoder, both projecting inputs into a shared embedding space. Given a dataset $\mathcal{D}$ of image-report pairs, we sample batches of images $\{\boldsymbol{I}_i\}$ and reports $\{\boldsymbol{T}_i\}$. The image encoder embeds each image $\boldsymbol{I}_i$ into the shared space, producing a feature vector $\boldsymbol{v}_i$. Similarly, the text encoder processes each report $\boldsymbol{T}_i$ and generates a corresponding feature vector $\boldsymbol{t}_i$ in the same space. CLIP optimizes a symmetric InfoNCE loss (Oord et al., 2018), treating paired image-report samples as positives and unpaired ones as negatives.

### 2.2 HIP-INFONCE: HOLISTIC IMAGE, PARTIAL-TEXT CONTRASTIVE LEARNING

Medical reports typically describe multiple findings per image. Instead of treating a report as indivisible, we generate multiple *masked text views* to represent partial descriptions.

For each report $\boldsymbol{T}_i$, we apply a *stochastic masking* operation:

$$\boldsymbol{T}_{i,k} = \text{MASK}_r^{\text{unif}}(\boldsymbol{T}_i),\tag{1}$$

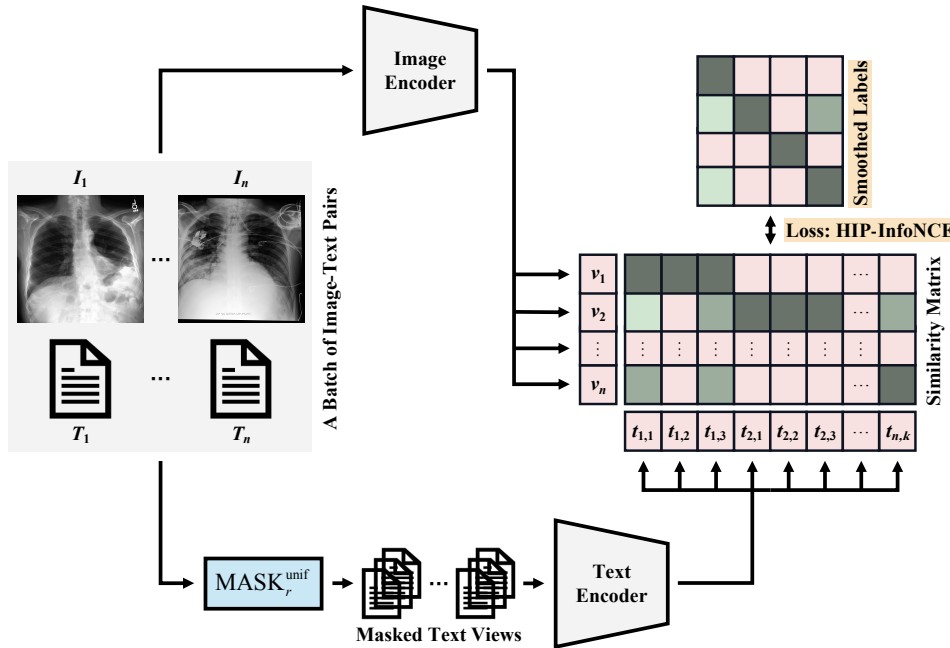

Figure 2: Overview of our CLIP-based framework for medical image-report pairs, featuring the proposed HIP-InfoNCE loss and text-aware label smoothing regularization.

where tokens are masked uniformly with ratio $r$, producing $K$ different masked text views. Let $\boldsymbol{t}_{i,k}$ be the embedding of $\boldsymbol{T}_{i,k}$. The HIP-InfoNCE loss is then:

$$
\mathcal{L}_{\texttt{HIP}}^{\texttt{v2t}} = -\sum_{i=1}^{N} \log \frac{\sum_{k=1}^{K} \exp(\boldsymbol{v}_i^{\top} \boldsymbol{t}_{i,k}/\tau)}{\sum_{j=1}^{N} \sum_{k=1}^{K} \exp(\boldsymbol{v}_i^{\top} \boldsymbol{t}_{j,k}/\tau)},
$$

$$
\mathcal{L}_{\texttt{HIP}}^{\texttt{t2v}} = -\sum_{i=1}^{N} \log \frac{\sum_{k=1}^{K} \exp(\boldsymbol{v}_i^{\top} \boldsymbol{t}_{i,k}/\tau)}{\sum_{j=1}^{N} \sum_{k=1}^{K} \exp(\boldsymbol{v}_j^{\top} \boldsymbol{t}_{i,k}/\tau)}, \tag{2}
$$

$$
\mathcal{L}_{\texttt{HIP}} = \tfrac{1}{2}(\mathcal{L}_{\texttt{HIP}}^{\texttt{v2t}} + \mathcal{L}_{\texttt{HIP}}^{\texttt{t2v}}),
$$

where $\tau$ is a learnable temperature parameter.

**Relation to Prior Work** Recent natural image-text alignment approaches employ multimodal large language models, such as ChatGPT, to augment captions, either rewriting captions for diverse views while still applying InfoNCE objectives (Fan et al., 2023; Liu et al., 2023b) or expanding short captions and applying sentence-level losses (Zheng et al., 2024). Our approach differs by: (1) operating on inherently long medical reports, using stochastic masking instead of sentence segmentation to avoid natural language processing (NLP) heuristics and domain expertise requirements; (2) capturing latent textual variations without being constrained to sentence boundaries, improving generalization. Empirically, HIP-InfoNCE outperforms sentence-based baselines, validating our design (cf. Section 4.5.2).

## 2.3 TEXT-AWARE LABEL SMOOTHING

As noted in Section 1, multiple images within a batch may share similar medical findings, making CLIP's strict one-to-one correspondence potentially suboptimal. To address this, we propose replacing hard labels with *text-aware soft labels*, extending the concept of label smoothing.

Traditional label smoothing (Szegedy et al., 2016) was introduced as a regularization technique to mitigate overfitting in classification networks. For a sample belonging to class $c$, the one-hot label

is replaced with a softened target $\boldsymbol{y} \in \mathbb{R}^C$ defined as:

$$\boldsymbol{y}[i] = \begin{cases} 1 - \epsilon & \text{if } i = c \,, \\ \epsilon/(C-1) & \text{otherwise} \,, \end{cases} \tag{3}$$

where $\epsilon$ is a small constant (e.g., 0.2) and $C$ the number of classes.

We extend this idea. While vanilla label smoothing uniformly distributes $\epsilon$ across negative terms, our text-aware label smoothing incorporates semantic information from medical reports. For a batch of $N$ image-report pairs, we define a soft label matrix $\boldsymbol{Y} \in \mathbb{R}^{N \times N}$ as:

$$\boldsymbol{Y}[i,j] = \begin{cases} 1 & \text{if } i = j \,, \\ \text{sim}(\boldsymbol{T}_i, \boldsymbol{T}_j) & \text{otherwise} \,, \end{cases} \tag{4}$$

where $\text{sim}(\cdot, \cdot)$ measures report similarity. We adopt BLEU-4 (Papineni et al., 2002) as our default metric, as it effectively captures n-gram overlaps in medical reports while balancing precision and recall. The resulting matrix undergoes row-wise normalization.

Thus, the modified InfoNCE loss becomes:

$$\mathcal{L}_{\text{TLS}}^{\text{v2t}} = -\sum_{i=1}^{N} \sum_{j=1}^{N} \boldsymbol{Y}[i,j] \log \frac{\exp(\boldsymbol{v}_i^\top \boldsymbol{t}_j / \tau)}{\sum_{l=1}^{N} \exp(\boldsymbol{v}_i^\top \boldsymbol{t}_l / \tau)} \,,$$

$$\mathcal{L}_{\text{TLS}}^{\text{t2v}} = -\sum_{i=1}^{N} \sum_{j=1}^{N} \boldsymbol{Y}[i,j] \log \frac{\exp(\boldsymbol{v}_j^\top \boldsymbol{t}_i / \tau)}{\sum_{l=1}^{N} \exp(\boldsymbol{v}_l^\top \boldsymbol{t}_i / \tau)} \,, \tag{5}$$

$$\mathcal{L}_{\text{TLS}} = \frac{1}{2}(\mathcal{L}_{\text{TLS}}^{\text{v2t}} + \mathcal{L}_{\text{TLS}}^{\text{t2v}}) \,.$$

**Choice of Similarity Metric**  While BLEU-4 serves as our primary similarity function, we investigate alternatives such as BLEU-2/6 and embedding-based metrics (e.g., BERTScore (Zhang et al., 2020)) in Section 4.5.3. BLEU-4 consistently yields the strongest retrieval performance, validating our choice.

## 2.4 Overall Loss

By integrating HIP-InfoNCE and text-aware label smoothing, we arrive at our final loss function for contrastive learning with medical image-report pairs:

$$\mathcal{L}_{\text{HIP-TLS}}^{\text{v2t}} = -\sum_{i=1}^{N} \sum_{j=1}^{N} \boldsymbol{Y}[i,j] \log \frac{\sum_{k=1}^{K} \exp(\boldsymbol{v}_i^\top \boldsymbol{t}_{j,k} / \tau)}{\sum_{l=1}^{N} \sum_{k=1}^{K} \exp(\boldsymbol{v}_i^\top \boldsymbol{t}_{l,k} / \tau)} \,,$$

$$\mathcal{L}_{\text{HIP-TLS}}^{\text{t2v}} = -\sum_{i=1}^{N} \sum_{j=1}^{N} \boldsymbol{Y}[i,j] \log \frac{\sum_{k=1}^{K} \exp(\boldsymbol{v}_j^\top \boldsymbol{t}_{i,k} / \tau)}{\sum_{l=1}^{N} \sum_{k=1}^{K} \exp(\boldsymbol{v}_l^\top \boldsymbol{t}_{i,k} / \tau)} \,, \tag{6}$$

$$\mathcal{L}_{\text{HIP-TLS}} = \frac{1}{2}(\mathcal{L}_{\text{HIP-TLS}}^{\text{v2t}} + \mathcal{L}_{\text{HIP-TLS}}^{\text{t2v}}) \,.$$

Figure 2 provides an overview of the proposed framework.

## 3 Related Work

### 3.1 Cross-Modal Retrieval

Early cross-modal retrieval methods align image and text representations by learning a shared embedding space. These approaches typically optimize ranking losses to match global image and text features (Faghri et al., 2017). With the remarkable success of CLIP (Radford et al., 2021) in open-domain image-text alignment, it becomes the dominant paradigm for cross-modal retrieval, inspiring numerous extensions. Architectural advances (Li et al., 2021; 2022; Jiang & Ye, 2023) explore strategies such as align-before-fusion (Li et al., 2021). On the data side, some methods like Fan et al.

(2023); Liu et al. (2023b); Zheng et al. (2024); Wang et al. (2025) improve model performance by augmenting image captions.

In the medical domain, CheXzero (Tiu et al., 2022) and ConVIRT (Zhang et al., 2022) align global image and text features, while GLoRIA (Huang et al., 2021), MGCA (Wang et al., 2022a), and LIMITR (Dawidowicz et al., 2023) leverage local visual features for more fine-grained alignment. CXR-CLIP (You et al., 2023) introduces consistency constraints through image and text augmentations. In addition, BioViL (Boecking et al., 2022), MaCo (Huang et al., 2024b), and MMCLIP (Wu et al., 2024) employ masked modeling strategies to learn cross-modal correspondence by predicting masked elements in images and texts. BioViL-T (Bannur et al., 2023) and Med-ST (Yang et al., 2024) further incorporate temporal context, modeling longitudinal dependencies across sequential chest X-rays and reports. MLIP (Li et al., 2024) and MedTrim (Ozturk et al., 2025) integrate clinical prior knowledge to enhance model performance. Despite these advances, the *holistic image versus multi-faceted report* property remains insufficiently explored and modeled.

### 3.2 LABEL SMOOTHING

Label smoothing replaces one-hot targets with softened distributions, assigning small non-zero probabilities to non-target classes (Szegedy et al., 2016). It mitigates overfitting in image classification tasks, and has been tailored to object detection through rotation angle label smoothing (ARS-DETR (Zeng et al., 2023)) and circular smoothing for arbitrary-oriented object detection (Yang & Yan, 2020). Similar extensions also benefit dense prediction tasks such as semantic segmentation (Park et al., 2023).

In the context of CLIP, SoftCLIP (Gao et al., 2024) constructs soft labels by leveraging similarities between local image regions and texts, requiring additional object detection models to extract region-of-interest (ROI) features and labels. Other works (Ko & Park, 2025) use cosine similarities between text embeddings as soft labels. MedCLIP (Wang et al., 2022b) employs additional diagnosis labels (in multi-hot encoding form), obtaining soft labels through dot products between encoded vectors. However, such labels have limited expressiveness and struggle to capture complex and nuanced semantics in medical reports, resulting in constrained effectiveness on downstream cross-modal retrieval tasks. In contrast, our method derives soft labels using semantic similarities of complete reports, more accurately reflecting semantic differences between cases and providing richer supervision signals.

## 4 EXPERIMENTS

### 4.1 DATASETS AND EVALUATION METRICS

We evaluate our approach on three datasets to assess both performance and generalization.

**MIMIC-CXR** (Johnson et al., 2019) is a large-scale chest X-ray dataset containing 377,110 images from 227,835 radiographic studies. Each study comprises a radiology report and at least one frontal image, with some including additional lateral views. We randomly sample 1,000 studies each for validation and test sets, with the remaining studies used as training data.

**CheXpert5x200** (Irvin et al., 2019) is a subset of CheXpert-v1.0 consisting of 1,000 chest X-rays (5 pathologies $\times$ 200 studies) with expert-annotated reports. This dataset provides a focused evaluation benchmark for cross-modal retrieval.

**IU X-ray** (Demner-Fushman et al., 2015) contains 7,470 chest X-rays from 3,955 radiology reports collected from Indiana University. Each report contains both findings and impression sections. Compared with MIMIC-CXR, its reports are shorter and often less detailed.

Since CheXpert5x200 and IU X-ray contain insufficient training samples for effective contrastive learning, we adopt a unified training strategy: all models are trained on the MIMIC-CXR training set, then evaluated on (1) the MIMIC-CXR test set for in-domain performance assessment and (2) CheXpert5x200 and IU X-ray for cross-domain generalization analysis.

We adopt standard retrieval metrics following You et al. (2023): Recall@K (R@K) and RSUM. R@K measures the percentage of queries where the correct match appears within the top-K retrieved

Table 1: Quantitative comparison of different retrieval methods on the MIMIC-CXR dataset. Results are shown for two backbone architectures.

| | | Image to Text | | | Text to Image | | | RSUM |
|---|---|---|---|---|---|---|---|---|
| | | R@1 | R@5 | R@10 | R@1 | R@5 | R@10 | |
| MGCA (Wang et al., 2022a) | RNet-50 | 26.9 | 50.6 | 62.4 | 23.3 | 49.1 | 62.2 | 274.5 |
| Med-ST (Yang et al., 2024) | | 24.2 | 51.8 | 65.5 | 24.6 | 53.1 | 64.5 | 283.7 |
| ConVIRT (Zhang et al., 2022) | | 28.4 | 53.4 | 63.3 | 28.6 | 52.1 | 64.1 | 289.9 |
| GLoRIA (Huang et al., 2021) | | 27.5 | 52.6 | 62.6 | 29.0 | 54.6 | 65.1 | 291.4 |
| CLIP (Radford et al., 2021) | | 28.0 | 53.7 | 65.2 | 28.4 | 53.2 | 64.6 | 293.1 |
| IRRA (Jiang & Ye, 2023) | | 32.1 | 54.7 | 66.2 | 33.3 | 55.5 | 66.5 | 308.3 |
| LIMITR (Dawidowicz et al., 2023) | | 30.2 | 57.4 | 66.9 | 31.2 | 57.4 | 65.9 | 309.0 |
| MedTrim (Ozturk et al., 2025) | | 34.2 | 58.6 | 68.7 | 35.2 | 59.3 | 67.8 | 323.8 |
| CXR-CLIP (You et al., 2023) | | 36.2 | 60.5 | 69.5 | 37.6 | 61.3 | 69.4 | 334.5 |
| Ours | | **36.7** | **63.0** | **71.4** | **39.7** | **61.9** | **72.1** | **344.8** |
| CLIP-Adaptor (Qin et al., 2024) | ViT | 24.8 | 50.5 | 60.4 | 23.7 | 49.6 | 61.8 | 270.8 |
| MGCA (Wang et al., 2022a) | | 26.5 | 49.5 | 61.8 | 25.6 | 49.9 | 61.6 | 274.9 |
| CLIP (Radford et al., 2021) | | 29.3 | 55.4 | 65.4 | 29.2 | 55.3 | 65.4 | 300.0 |
| MaCo (Huang et al., 2024b) | | 32.1 | 55.8 | 66.6 | 33.0 | 59.2 | 67.5 | 314.2 |
| IRRA (Jiang & Ye, 2023) | | 34.4 | 57.9 | 65.6 | 34.5 | 60.5 | 66.5 | 319.4 |
| MedTrim (Ozturk et al., 2025) | | 36.2 | 60.4 | 70.7 | 37.5 | 61.8 | 68.1 | 334.7 |
| CXR-CLIP (You et al., 2023) | | 39.2 | 61.4 | 69.7 | 38.5 | 61.2 | 69.4 | 339.4 |
| Ours | | **41.4** | **63.4** | **72.5** | **40.2** | **65.2** | **72.3** | **355.0** |

results. We report R@K for K = 1, 5, 10 to provide a comprehensive assessment. RSUM, computed as the sum of all R@K values, serves as an aggregate measure of overall retrieval performance.

## 4.2 IMPLEMENTATION DETAILS

All images are resized to 256×256 pixels and then center-cropped to 224×224. Reports are truncated or padded to 150 tokens. We use a 512-dimensional embedding space. We employ ResNet-50 and vision Transformer as our vision backbone. For ResNet-50, visual features are extracted from the fourth residual block, while for vision Transformer, we use the [CLS] token representation. Both are followed by a linear projection into the embedding space. For text encoding, we adopt BERT, extracting text representations from the [EOS] token and projecting them to the embedding space through a linear transformation. For HIP-InfoNCE, the sampling number $K$ and masking ratio $r$ are set to 4 and 0.3, respectively, based on empirical validation (see Section 4.5.1). The learnable temperature parameter $\tau$ is initialized to 0.07. Training uses the AdamW (Loshchilov & Hutter, 2017) optimizer with learning rate 5e-5, weight decay 1e-5, and batch size 128. All models are trained for 80 epochs on two NVIDIA GeForce RTX 4090 GPUs. Training typically converges within 60-70 epochs. Our implementation is in PyTorch.

## 4.3 COMPARISON WITH STATE-OF-THE-ART METHODS

Table 1 presents a comparison with current state-of-the-art approaches. We compare against both medical domain-specific approaches (MedTrim (Ozturk et al., 2025), Med-ST (Yang et al., 2024), MaCo (Huang et al., 2024b), CLIP-Adaptor (Qin et al., 2024), CXR-CLIP (You et al., 2023), LIMITR (Dawidowicz et al., 2023), MGCA (Wang et al., 2022a), GLoRIA (Huang et al., 2021), and ConVIRT (Zhang et al., 2022)) and general-domain methods (IRRA (Jiang & Ye, 2023) and CLIP (Radford et al., 2021)).

Note that several competing methods leverage additional information: Med-ST utilizes temporal cues, while LIMITR and CXR-CLIP incorporate lateral views. CXR-CLIP further benefits from extensive data augmentation and self-supervised learning.

Our method achieves state-of-the-art performance with an RSUM of 355.0, surpassing CXR-CLIP by 15.6 points. Gains over LIMITR, Med-ST, and MGCA are 46.0, 71.3, and 80.1 points, respectively. Notably, our method achieves superior performance across all individual metrics (R@1, R@5, R@10) for both image-to-text and text-to-image retrieval directions without relying on additional data sources or augmentation strategies.

Table 2: Ablation study showing individual contributions of HIP-InfoNCE (HIP) and text-aware label smoothing (TLS).

| HIP | TLS | Image to Text | | | Text to Image | | | RSUM |
|---|---|---|---|---|---|---|---|---|
| | | R@1 | R@5 | R@10 | R@1 | R@5 | R@10 | |
| - | - | 32.1 | 57.7 | 67.2 | 31.7 | 57.7 | 67.2 | 313.6 |
| ✓ | - | 36.9 | 61.6 | 71.6 | 36.6 | 61.9 | 70.7 | 339.3 |
| - | ✓ | 36.6 | 61.6 | 70.8 | 38.2 | 61.0 | 69.1 | 337.3 |
| ✓ | ✓ | **41.4** | **63.4** | **72.5** | **40.2** | **65.2** | **72.3** | **355.0** |

## 4.4 ABLATION STUDIES

Table 2 presents systematic ablation studies examining the contribution of each proposed component. Starting from the baseline InfoNCE loss, we observe that:

**HIP-InfoNCE alone** provides consistent improvements (+25.7 RSUM) across both retrieval directions. This validates our hypothesis that modeling the holistic-to-partial alignment through masked text views better captures the multi-faceted nature of medical reports.

**Text-aware label smoothing alone** yields significant improvements (+23.7 RSUM), with particularly strong gains in text-to-image retrieval (+6.5 R@1), confirming that its soft label formulation enhances contrastive learning in the medical domain.

**Combined approach** achieves optimal performance (+41.4 RSUM), demonstrating that both contributions are complementary.

## 4.5 MODEL ANALYSIS

### 4.5.1 IMPACT OF HYPERPARAMETERS

We begin by examining the influence of two key hyperparameters: the number of masked text views $K$ and the masking ratio $r$. Figure 3 presents a radar chart visualization of performance across different parameter combinations. Results demonstrate that $r = 0.3$ achieves the best balance between semantic preservation and information diversity, as evidenced by the largest enclosed area in the radar chart. Masking ratios exceeding 0.5 lead to convergence instability due to excessive information loss. For $K$, we observe performance improvements as more masked text views are incorporated. However, marginal gains diminish beyond $K = 4$. The optimal configuration occurs at $K = 4$ and $r = 0.3$, where our model achieves the strongest retrieval performance.

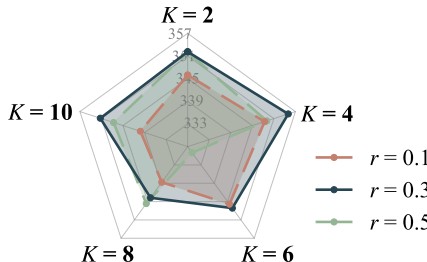

Figure 3: Hyperparameter analysis using a radar chart visualization. Performance is measured using RSUM.

### 4.5.2 IMAGE-TEXT ALIGNMENT STRATEGY ANALYSIS

Table 3 compares different strategies for constructing text views and aligning images with reports. Starting from the baseline in Section 4.4 (RSUM 313.6), incorporating stochastic masking purely as data augmentation yields a clear gain (+13.3 RSUM), confirming that stochastic masking enhances representation learning through semantic variation, albeit with limited effect. In contrast, randomly sampling a single sentence per report as data augmentation (cf. Fan et al. (2023); Liu et al. (2023b)) degrades performance (RSUM 314.7). Randomly sampling multiple sentences and applying a one-to-many alignment loss (Zheng et al., 2024) yields only a marginal improvement (317.3). Both approaches fragment clinical narratives into disjoint sentences, disrupting semantic continuity and discarding contextual dependencies crucial for effective retrieval.

Our HIP-InfoNCE significantly outperforms all alternatives, achieving 339.3 RSUM (+25.7). Specifically, image-to-text R@1 improves from 32.1 to 36.9, while text-to-image R@5 increases from 57.7 to 61.9. This advantage stems from two key design principles: (1) stochastic masking diversifies textual perspectives while preserving semantic coherence, and (2) multi-view alignment

Table 3: Analysis of different image-text alignment strategies. Stoch. Mask. refers to stochastic masking, Sent. Samp. to sentence sampling, Data Aug. to data augmentation, and 1-to-M. Align. to one-to-many alignment.

| | Image to Text | | | Text to Image | | | RSUM |
|---|---|---|---|---|---|---|---|
| | R@1 | R@5 | R@10 | R@1 | R@5 | R@10 | |
| Baseline | 32.1 | 57.7 | 67.2 | 31.7 | 57.7 | 67.2 | 313.6 |
| Stoch. Mask. as Data Aug. | 35.2 | 57.9 | 69.6 | 35.7 | 60.4 | 68.1 | 326.9 |
| Sent. Samp. as Data Aug. | 31.5 | 57.3 | 68.3 | 30.5 | 58.4 | 68.7 | 314.7 |
| Sent. Samp. & 1-to-M. Align. | 32.0 | 58.2 | 68.9 | 31.7 | 58.9 | 67.6 | 317.3 |
| HIP-InfoNCE | **36.9** | **61.6** | **71.6** | **36.6** | **61.9** | **70.7** | **339.3** |

Table 4: Comparison of similarity metrics for text-aware label smoothing. Van. Lab. Smooth. denotes vanilla label smoothing. BERTScore uses BiomedBERT, as it achieves the best performance; results with other pre-trained language models are provided in Appendix C.

| | Image to Text | | | Text to Image | | | RSUM |
|---|---|---|---|---|---|---|---|
| | R@1 | R@5 | R@10 | R@1 | R@5 | R@10 | |
| Van. Lab. Smooth. | 37.8 | 62.6 | 71.3 | 38.3 | 62.8 | 71.2 | 344.0 |
| BLEU-2 | 40.7 | 61.5 | 69.6 | 39.7 | 62.6 | 71.1 | 345.2 |
| BLEU-4 | **41.4** | **63.4** | **72.5** | 40.2 | **65.2** | **72.3** | **355.0** |
| BLEU-6 | 39.9 | 63.1 | 71.6 | **40.9** | 63.3 | 72.2 | 351.0 |
| Rouge-L | 38.3 | 63.0 | 72.2 | 38.6 | 62.9 | 71.4 | 346.4 |
| BERTScore | 40.4 | 63.0 | 70.1 | 39.8 | 63.7 | 71.1 | 348.1 |

effectively captures the inherently multi-faceted nature of chest X-ray descriptions. Importantly, our approach requires no manual parsing or domain-specific knowledge, making the approach both effective and practical.

### 4.5.3 TEXT-AWARE LABEL SMOOTHING

We next evaluate different similarity metrics for constructing text-aware soft labels (see Table 4). Vanilla label smoothing (Szegedy et al., 2016) performs poorly (RSUM 344.0), as uniform smoothing fails to leverage semantic relationships between medical cases. Rouge-L (Lin, 2004) and low-order n-gram BLEU (e.g., BLEU-2) (Papineni et al., 2002) yield modest gains. Metrics with stronger semantic priors perform better; for example, BERTScore$_{BiomedBERT}$ (Gu et al., 2020) leverages domain-specific contextual embeddings to capture deeper similarity. Interestingly, higher-order BLEU proves most effective: BLEU-6 captures long-span coherence, while BLEU-4 strikes the best balance between precision and recall, achieving the highest RSUM (355.0). These results confirm that text-aware label smoothing with appropriate similarity metrics strengthens representation learning.

### 4.5.4 CROSS-DOMAIN GENERALIZATION

We further validate the zero-shot generalization capability of our method on CheXpert5x200 and IU X-ray (Table 5). All models are trained exclusively on MIMIC-CXR and tested without fine-tuning.

**CheXpert5x200 results.** With ResNet-50, we achieve the highest RSUM of 99.9, excelling in image-to-text retrieval (R@1: 6.7, R@5: 17.7). Using vision Transformer, our method reaches an RSUM of 114.0, significantly surpassing CXR-CLIP (102.6) and MedTrim (106.1), with superior performance in 5 out of 6 metrics.

**IU X-ray results.** Despite a domain shift toward shorter, less detailed reports, our method maintains strong performance. With ResNet-50, we obtain RSUM 64.1, leading in text-to-image R@5/R@10 and image-to-text R@10. With vision Transformer, RSUM reaches 63.8, achieving the best text-to-image R@5/R@10 and image-to-text R@1/R@10.

The consistent superiority across diverse datasets demonstrates the robustness and generalizability of our design.

| Image | Ours | CXR-CLIP | MedTrim | IRRA | MaCo | CLIP |
|---|---|---|---|---|---|---|

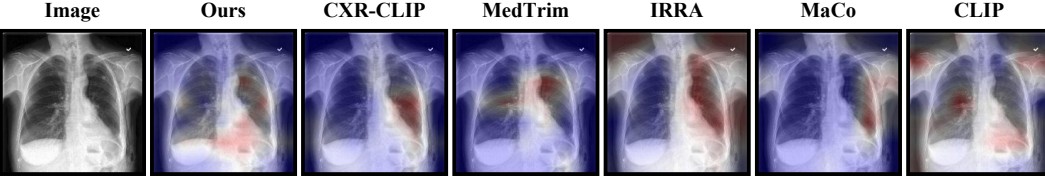

**Text:** *The air component of the moderate multiloculated left hydro pneumothorax, has grown larger. Small right pleural effusion unchanged. Moderate subcutaneous emphysema in the left chest wall is stable. Hyperinflation reflects severe emphysema. Right lung clear of focal abnormalities. The roughly 5.5 cm wide round opacity projecting over the left paraspinal region has increased since the earliest post surgical radiograph, and probably. On a subsequent chest radiograph available the time of this review, the abnormality has resolved, and it presumably was due to an unusual configuration of atelectasis.*

Figure 4: Attention visualization comparison using Grad-CAM. Our method generates more distributed attention patterns that align with multiple regions described in the text.

Table 5: Cross-domain generalization results on CheXpert5x200 and IU X-ray. All models are trained on MIMIC-CXR and evaluated zero-shot. Results are reported for both ResNet-50 (top) and vision Transformer (bottom) backbones.

| | CheXpert5x200 | | | | | | | IU X-ray | | | | | | |
|---|---|---|---|---|---|---|---|---|---|---|---|---|---|---|
| | Image to Text | | | Text to Image | | | RSUM | Image to Text | | | Text to Image | | | RSUM |
| | R@1 | R@5 | R@10 | R@1 | R@5 | R@10 | | R@1 | R@5 | R@10 | R@1 | R@5 | R@10 | |
| MGCA | 4.1 | 9.8 | 15.1 | 3.3 | 10.4 | 15.7 | 58.4 | 2.4 | 8.2 | 11.6 | 3.0 | 7.9 | 12.2 | 45.3 |
| Med-ST | 2.9 | 9.5 | 13.8 | 3.3 | 10.3 | 15.6 | 55.4 | 2.2 | 7.7 | 12.0 | 2.0 | 7.9 | 13.1 | 44.9 |
| ConVIRT | 3.9 | 12.9 | 18.0 | 4.7 | 13.4 | 21.6 | 74.5 | **4.8** | 9.2 | 12.5 | 4.6 | 11.0 | 13.9 | 56.0 |
| GLoRIA | 3.1 | 11.0 | 16.5 | 4.9 | 13.0 | 20.1 | 68.6 | 2.3 | 6.7 | 10.9 | 5.1 | 10.8 | 13.2 | 49.0 |
| CLIP | 4.5 | 13.5 | 20.2 | 5.7 | 15.1 | 21.3 | 80.3 | 3.4 | 9.6 | 13.9 | 4.8 | 10.6 | 14.1 | 56.4 |
| IRRA | 4.3 | 13.1 | 19.3 | 5.3 | 12.9 | 20.9 | 75.8 | 4.2 | 8.2 | 11.8 | **5.9** | 10.6 | 14.1 | 54.8 |
| LIMITR | 5.6 | 15.1 | 24.8 | 5.7 | 18.4 | **26.5** | 96.1 | 4.4 | 10.0 | 14.0 | 5.0 | 11.5 | 14.7 | 59.6 |
| MedTrim | 6.2 | 16.2 | 23.8 | 5.6 | 18.6 | 24.9 | 95.3 | 4.2 | 10.4 | 13.2 | 4.1 | 10.3 | 15.5 | 57.7 |
| CXR-CLIP | 6.0 | 15.8 | 22.6 | **7.1** | **19.4** | 25.5 | 96.4 | 4.0 | **11.0** | 14.1 | 4.8 | 11.4 | 14.3 | 59.6 |
| Ours | **6.7** | **17.7** | **24.8** | 6.2 | 18.5 | 26.0 | **99.9** | 4.4 | 10.8 | **14.4** | 5.2 | **11.8** | **17.5** | **64.1** |
| CLIP-Adapt. | 5.7 | 16.8 | 23.2 | 5.1 | 17.0 | 24.3 | 92.1 | 3.5 | 10.6 | 13.9 | **5.2** | 11.6 | 14.8 | 59.6 |
| MGCA | 5.5 | 14.4 | 18.9 | 4.2 | 12.6 | 18.9 | 74.5 | 2.7 | 8.2 | 10.9 | 4.3 | 10.1 | 13.5 | 49.7 |
| CLIP | 6.6 | 16.7 | 23.5 | 6.5 | 16.5 | 24.4 | 94.2 | 3.2 | 8.2 | 11.6 | 4.0 | 10.5 | 14.0 | 51.5 |
| MaCo | 6.5 | 15.7 | 22.8 | 5.6 | 17.2 | 26.9 | 94.7 | 3.1 | 8.1 | 11.2 | 4.4 | 11.3 | 15.1 | 53.2 |
| IRRA | 6.5 | 16.3 | 24.7 | 6.9 | 17.6 | 25.0 | 97.0 | 3.2 | 9.7 | 10.8 | 4.8 | 11.9 | 15.1 | 55.5 |
| MedTrim | **7.5** | 19.2 | 27.3 | 7.2 | 18.6 | 26.3 | 106.1 | 4.2 | 9.2 | 12.8 | 4.7 | 10.8 | 15.0 | 56.7 |
| CXR-CLIP | 6.9 | 18.2 | 26.9 | 6.7 | 18.1 | 25.8 | 102.6 | 3.9 | **11.2** | 14.3 | 4.2 | 10.6 | 14.0 | 58.2 |
| Ours | 7.4 | **20.9** | **28.5** | **9.5** | **19.8** | **27.9** | **114.0** | **5.4** | 10.9 | **14.3** | 5.0 | **12.2** | **16.0** | **63.8** |

### 4.5.5 QUALITATIVE ANALYSIS

Qualitative analysis through attention visualization using Grad-CAM (Selvaraju et al., 2017) (Figure 4) demonstrates our model's ability to align image regions with textual descriptions. Our method produces more distributed and fine-grained attention patterns that cover multiple key medical findings mentioned in the report (e.g., right pleural effusion, subcutaneous emphysema, and opacity projecting). In contrast, other methods exhibit overly concentrated attention on single image regions, producing coarse heatmaps. This semantic alignment advantage stems from our alignment mechanism, which better reflects the multi-faceted structure of medical reports. We show qualitative retrieval examples in Appendix D.

## 5 CONCLUSION

In this paper, we present a framework for medical image-report retrieval that addresses domain-specific limitations of existing CLIP-based methods. Our key contributions—HIP-InfoNCE loss and text-aware label smoothing—better capture complex image-report relationships. Extensive experiments show state-of-the-art performance, highlighting the importance of incorporating domain-specific properties into the design of retrieval systems for medical applications. Future work will investigate the application of our CLIP framework to tasks beyond cross-modal retrieval.

ETHICS STATEMENT

The authors acknowledge that this work adheres to the ICLR Code of Ethics.

REPRODUCIBILITY STATEMENT

Code to reproduce all experiments is available at `https://anonymous.4open.science/r/CLIP-HIP-TLS-A54B/`.

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

# APPENDIX

## A  USE OF LARGE LANGUAGE MODELS

Large language models were used solely for light editing tasks including grammar correction, spelling checks, and minor phrasing improvements to enhance clarity and concision.

## B  PSEUDO-CODE FOR CROSS-MODAL RETRIEVAL

Algorithm 1 and Algorithm 2 present the pseudo-code of cross-modal retrieval with our model. Specifically, image and text embeddings are extracted by the image encoder and text encoder, respectively, followed by normalization. Similarity scores are then computed via dot products, and the Top-$k$ candidates with their scores are returned.

---
**Algorithm 1** Pseudo-code of image-to-text retrieval.

---
```python
def image_to_text_retrieval(I, T):
    v = image_encoder(I)
    t = text_encoder(T)

    v = norm(v, dim=-1)
    t = norm(t, dim=-1)

    sims = matmul(v, t.T)
    scores, indices = topk(sims, k, dim=-1)

    return T[indices], scores
```
---

---
**Algorithm 2** Pseudo-code of text-to-image retrieval.

---
```python
def text_to_image_retrieval(T, I):
    t = text_encoder(T)
    v = image_encoder(I)

    t = norm(t, dim=-1)
    v = norm(v, dim=-1)

    sims = matmul(t, v.T)
    scores, indices = topk(sims, k, dim=-1)

    return I[indices], scores
```
---

## C BERTScore

We examine the performance of different pre-trained language models within the BERTScore framework when used as the similarity metric in our label smoothing method. Table 6 reports results. Overall, the performance differences are minor. We adopt BiomedBERT as the default pre-trained model for BERTScore in our experiments.

Table 6: Comparison of different pre-trained language models within BERTScore.

| | Image to Text | | | Text to Image | | | RSUM |
|---|---|---|---|---|---|---|---|
| | R@1 | R@5 | R@10 | R@1 | R@5 | R@10 | |
| Medical-NER (Campillos-Llanos et al., 2021) | 39.8 | 62.4 | **71.5** | 40.1 | 62.6 | 70.7 | 347.1 |
| DistilBERT (Sanh et al., 2019) | **40.9** | 61.8 | 70.6 | 40.5 | 62.7 | 70.6 | 347.1 |
| BioClinicalBERT (Alsentzer et al., 2019) | 38.3 | **63.6** | 70.6 | 39.3 | 63.5 | **72.0** | 347.3 |
| BiomedBERT (Gu et al., 2020) | 40.4 | 63.0 | 70.1 | 39.8 | **63.7** | 71.1 | **348.1** |
| DeBERTa (He et al., 2023) | 38.7 | 61.5 | 70.6 | 39.5 | 62.9 | 70.6 | 343.8 |
| MedCPT (Jin et al., 2023) | 39.9 | 62.7 | 69.8 | **40.7** | 63.4 | 70.4 | 346.9 |

## D Retrieval Examples

Figure 6 presents both text-to-image and image-to-text results with our model. Figures 7, 8, and 9 show qualitative retrieval examples on MIMIC-CXR, CheXpert5x200, and IU-Xray, comparing our method with baselines.

## E Hyperparameter Analysis

Table 7 provides a more detailed hyperparameter analysis, including all evaluation metrics for completeness.

Table 7: Impact of hyperparameters $K$ (number of masked text views) and $r$ (masking ratio) on retrieval performance.

| $K$ | $r$ | Image to Text | | | Text to Image | | | RSUM |
|---|---|---|---|---|---|---|---|---|
| | | R@1 | R@5 | R@10 | R@1 | R@5 | R@10 | |
| 2 | 0.1 | 40.1 | 62.0 | 70.8 | 39.7 | 62.4 | 70.9 | 345.9 |
| | 0.3 | 41.5 | **63.8** | 71.5 | 39.7 | 64.2 | 71.4 | 352.1 |
| | 0.5 | 41.1 | 63.1 | 72.0 | **40.9** | 63.4 | 72.1 | 352.6 |
| 4 | 0.1 | 38.9 | 63.4 | 71.0 | 39.7 | 64.3 | 71.1 | 348.4 |
| | 0.3 | **41.4** | 63.4 | **72.5** | 40.2 | **65.2** | 72.3 | **355.0** |
| | 0.5 | 39.1 | 62.4 | 71.9 | 38.6 | 64.3 | **72.7** | 349.0 |
| 6 | 0.1 | 39.0 | 63.1 | 70.2 | 40.6 | 62.6 | 70.3 | 345.8 |
| | 0.3 | 38.5 | 62.8 | 71.2 | 39.1 | 64.5 | 71.0 | 347.1 |
| | 0.5 | 36.3 | 59.7 | 68.1 | 36.7 | 59.5 | 68.6 | 328.9 |
| 8 | 0.1 | 37.8 | 62.6 | 69.0 | 36.7 | 63.2 | 69.3 | 338.6 |
| | 0.3 | 38.5 | 62.4 | 71.4 | 38.7 | 61.9 | 70.8 | 343.7 |
| | 0.5 | 36.7 | 63.7 | 70.6 | 39.7 | 63.2 | 71.7 | 345.6 |
| 10 | 0.1 | 40.6 | 60.9 | 69.8 | 39.2 | 61.3 | 68.3 | 340.1 |
| | 0.3 | 39.8 | 63.6 | 71.3 | 39.6 | 64.3 | 72.7 | 351.3 |
| | 0.5 | 40.3 | 63.4 | 71.2 | 39.4 | 62.1 | 71.2 | 347.6 |

## F Word Attention Visualization

Figure 5 visualizes the average multi-head attention from the last layer of our model's text encoder. In particular, it highlights the attention from the [EOS] token (used to aggregate semantic features) to each input word. The importance of each word is determined by the maximum attention weight across its sub-tokens.

The heatmap shows that the model assigns high attention to clinically critical terms. For example, pathological words such as *pneumothorax*, *emphysema*, and *hyperinflation*, as well as anatomical

words such as *left chest* and *thorax*, receive significantly higher attention. These terms correspond to the primary findings and locations emphasized in clinical diagnosis, suggesting that the model effectively captures key diagnostic cues. In contrast, functional or less informative words obtain lower attention, indicating the model's ability to filter irrelevant information.

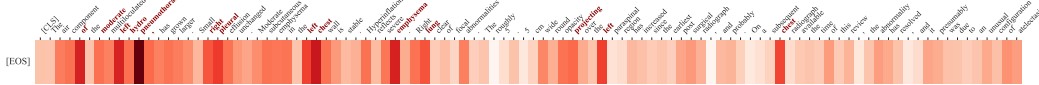

Figure 5: Word-level attention visualization. The model assigns high attention to key pathological and anatomical terms, indicating effective capture of diagnostic cues.

**Query**

*Left pigtail catheter is in place. Subcutaneous air appears to be minimally decreased. For 's left posterior aspect 3 fracture is unchanged. There is no pneumothorax. Lungs are overall clear. Small left pleural effusion is unchanged. The additional rib fractures also noted on the current examination.*

**Ranks**

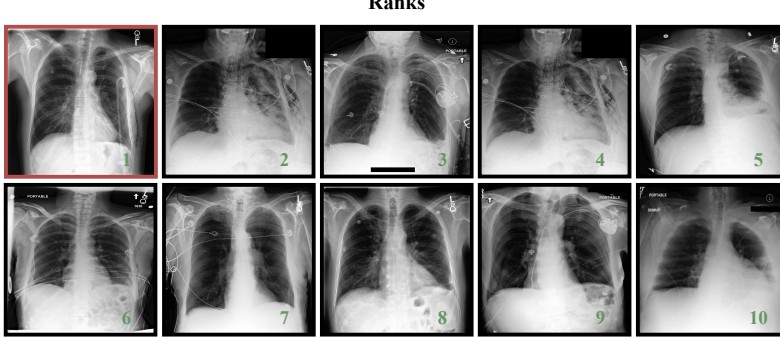

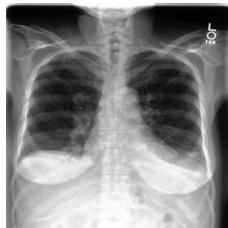

*Right pleural effusion has decreased in size following thoracentesis with an apparent tiny right apical pneumothorax and small residual pleural effusion. Small to moderate left pleural effusion is similar to the prior radiograph…* 1

*Signs of mild fluid overload have decreased in severity. Better seen than on the frontal radiograph, the lateral radiograph shows a moderate left and a small right pleural effusion…* 2

*As compared to the previous radiograph, the pre-existing right pleural effusion has slightly decreased, but the pre-existing left pleural effusion has slightly increased in extent…* 3

*Compared to chest radiograph, cardiomediastinal contours are stable. Substantial bibasilar atelectasis has slightly worsened accompanied by persistent small bilateral pleural effusions.* 4

*The pre-existing small bilateral pleural effusions are constant. Slightly increased in extent and severity is the known platelike atelectasis at the right lung bases…* 5

Figure 6: Retrieval results with our model. Top: text-to-image retrieval with query text and top-10 retrieved images. Bottom: image-to-text retrieval with query image and top-5 retrieved reports.

**Query:** *Left pigtail catheter is in place. Subcutaneous air appears to be minimally decreased. For 's left posterior aspect 3 fracture is unchanged. There is no pneumothorax. Lungs are overall clear. Small left pleural effusion is unchanged. The additional rib fractures also noted on the current examination.*

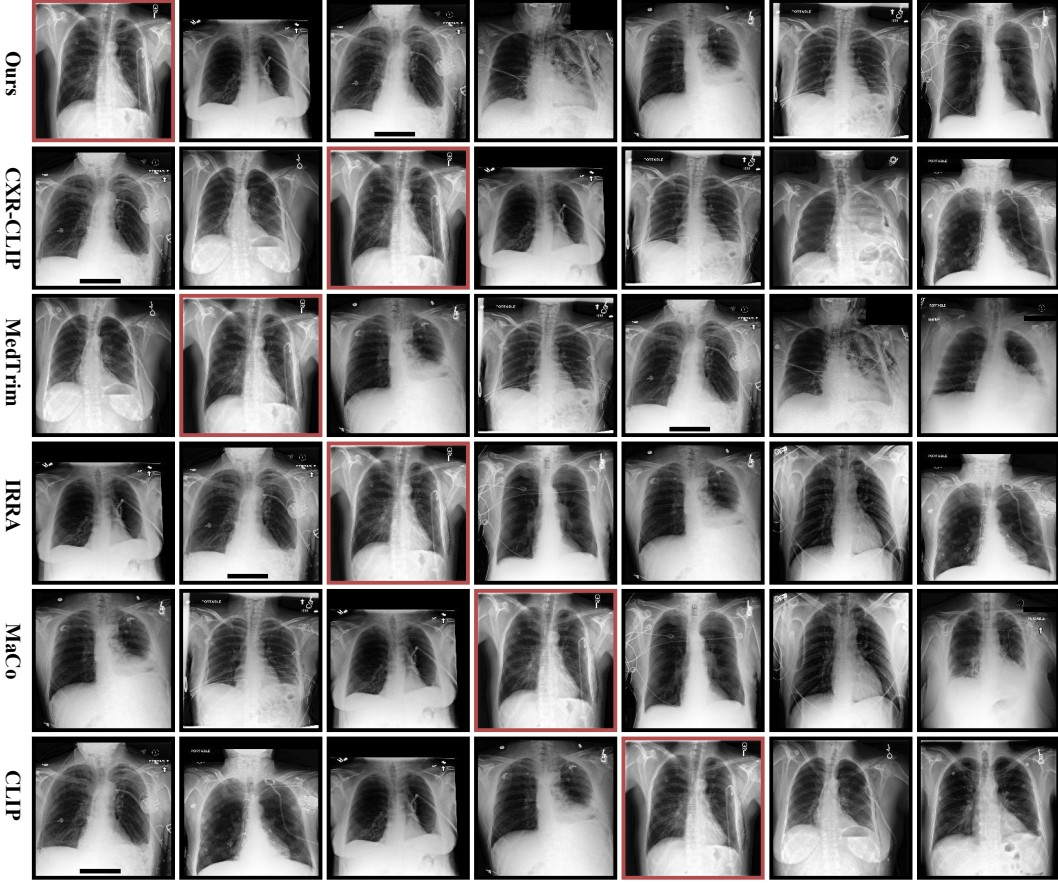

Figure 7: Retrieval examples on MIMIC-CXR, comparing our method with competitors.

**Query:** *As before there is partial left upper lobe collapse with mild mediastinal shift the right lung remains clear no pneumothorax or pleural effusions the heart size is normal.*

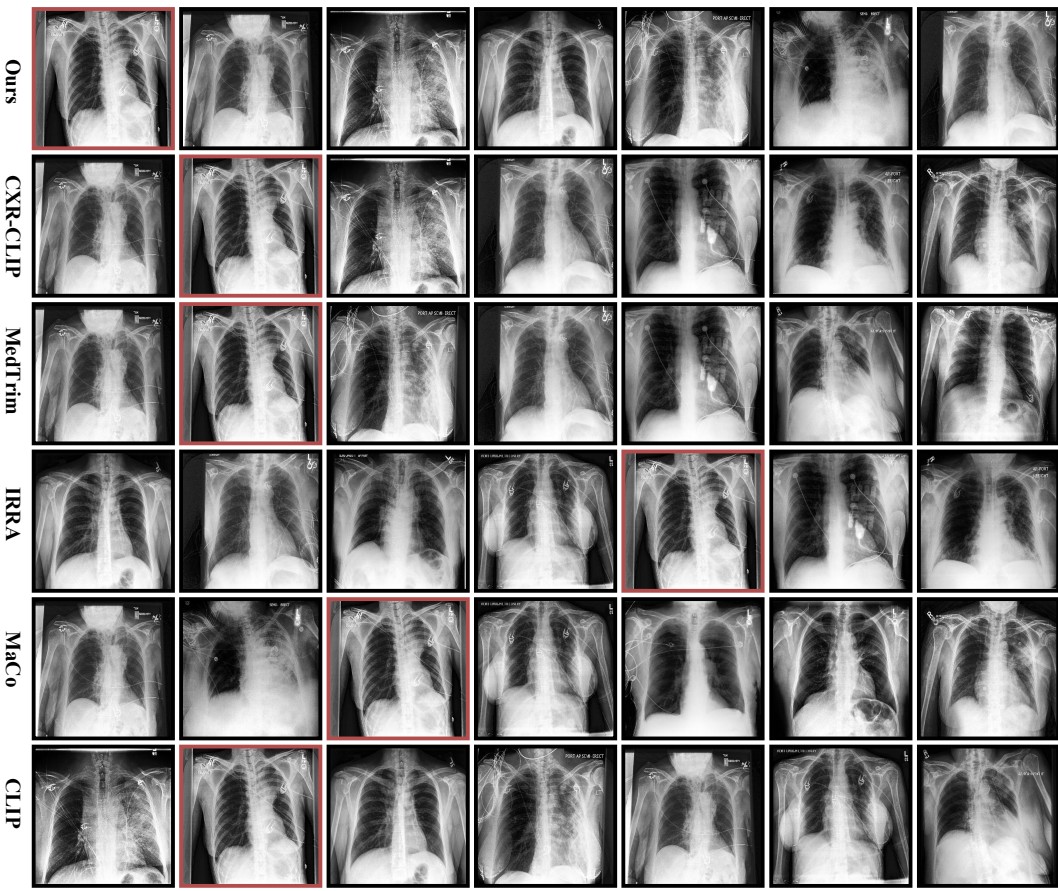

Figure 8: Retrieval examples on CheXpert5x200, comparing our method with competitors.

**Query:** *Small left pleural effusion stable mild cardiomegaly stable cardiomediastinal contour is no pneumothorax or significant pulmonary edema small left pleural effusion no focal lung consolidation mildly low lung volumes.*

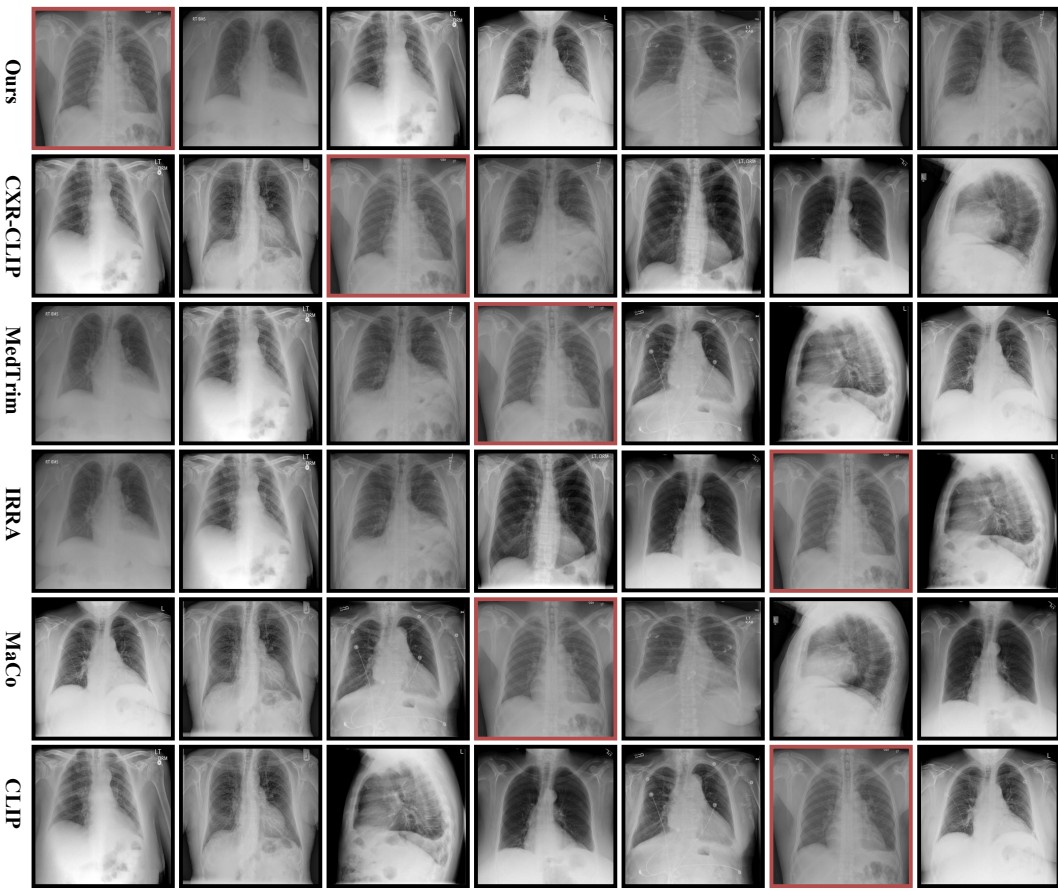

Figure 9: Retrieval examples on IU X-ray, comparing our method with competitors.

