# OpenReview forum: "Rethinking Contrastive Language-Image Pre-Training for Medical Cross-Modal Retrieval: Beyond One-to-One Correspondence"
_ICLR.cc/2026/Conference — ICLR 2026 Conference Withdrawn Submission_

### Official Review · Reviewer_wfuj · 2025-10-28

**Soundness:** 3
**Presentation:** 3
**Contribution:** 2
**Rating:** 4
**Confidence:** 4

**Summary:**

This paper addresses medical image-report retrieval by proposing two modifications to CLIP: HIP-InfoNCE for holistic-to-multi-faceted alignment using stochastic text masking, and text-aware label smoothing based on semantic similarity between reports. The methods achieve strong results on MIMIC-CXR, CheXpert5x200, and IU X-ray datasets with RSUM of 355.0, outperforming previous work by 15.6 points.

**Strengths:**

1. The paper identifies a genuine domain-specific challenge in medical imaging where single holistic images correspond to multi-faceted textual descriptions, which is well-motivated through empirical analysis in Figure 1.

2. The experimental evaluation is comprehensive and thorough, covering three datasets with extensive ablation studies, cross-domain generalization tests, and attention visualizations that provide interpretability.

3. The proposed methods achieve impressive state-of-the-art performance with significant improvements over strong baselines including CXR-CLIP, demonstrating the effectiveness of the approach.

**Weaknesses:**

1. The core contribution of text-aware label smoothing closely resembles MedCLIP's semantic matching loss, and the paper fails to adequately differentiate or demonstrate meaningful improvements over this well-established prior work.

2. Unlike recent state-of-the-art work like RadIR which constructs multi-grained similarity annotations from reports in a principled way, this paper's stochastic masking approach is ad-hoc and lacks clear theoretical or empirical justification for why random token masking should preserve semantic coherence better than natural linguistic boundaries.

3. The use of BLEU-4 for measuring report similarity is overly simplistic and inappropriate for medical text, as it cannot capture clinical semantics or distinguish between reports with opposite clinical meanings that share surface-level wording, whereas recent work uses sophisticated medical language models like RaTEScore for proper semantic understanding.

4. The paper operates only at the global image-report level without addressing fine-grained retrieval needs, while state-of-the-art systems like RadIR enable anatomy-conditioned retrieval that better aligns with clinical workflows where radiologists focus on specific anatomical regions.

5. The experimental validation is limited to retrieval metrics on existing datasets without demonstrating clinical utility, generalization to new domains, or downstream task performance, whereas strong papers contribute new datasets or benchmarks that advance the field beyond incremental metric improvements.

6. The computational cost of K=4 text encodings per sample is substantial but completely unanalyzed, and the marginal improvements (Table 3: 339.3 vs 317.3) do not clearly justify this overhead compared to more efficient approaches.

**Questions:**

1. Have you considered using sentence-level or finding-level segmentation instead of random token masking, and can you provide linguistic or clinical analysis showing why stochastic masking better preserves semantic coherence?

2. Have you evaluated the learned representations on downstream tasks such as disease classification or report generation to assess whether improved retrieval translates to better clinical utility?

---

### Official Review · Reviewer_cGtb · 2025-10-30

**Soundness:** 2
**Presentation:** 3
**Contribution:** 2
**Rating:** 2
**Confidence:** 4

**Summary:**

This paper addresses the issues of image-text granularity mismatch and semantic overlap between cases in the medical image-report cross-modal retrieval task. It proposes a novel contrastive learning framework, including the HIP-InfoNCE loss function and a text-aware label smoothing mechanism. Extensive experiments demonstrate that the proposed method achieves significant improvements over existing approaches and reaches state-of-the-art performance.

**Strengths:**

Originality. While the components of this paper build upon existing techniques such as CLIP, multi-view contrastive learning, and label smoothing, their combination for the specific goal of medical image-report retrieval is novel and well-motivated.

Quality. The method is technically sound, the model architecture is reasonably chosen, ablation experiments isolate the contribution of each module, and the comparative experiments include recent strong baselines.

Clarity. The paper is well-written, with corresponding formulas for the loss functions, making it easy to understand. The descriptions of experimental details such as training data and hyperparameters are sufficiently transparent, and replication appears feasible.

Significance. The proposed method achieves a approximate 20-point improvement in RSUM on the MIMIC-CXR dataset compared to existing SOTA results in 2025. Therefore, the proposed chest radiograph retrieval and report-based case retrieval are not merely academic benchmarks; they can be linked to practical use cases such as clinical decision support, thus possessing strong application value.

**Weaknesses:**

1. The "analysis challenge" in contribution point 1 is not a substantial technical contribution; it merely lays the groundwork for the latter two methods and does not constitute an independent achievement.

2. In the general vision-language domain, some researchers use "random cropping" augmentation for multi-view alignment, essentially treating multiple views of the same semantic object as positive samples. Therefore, this work seems essentially no different from sentence-level alignment and local alignment work;      this paper appears more like text data augmentation than a theoretical innovation.

3. The causal chain between the motivation of "semantic overlap" and the proposed "text-aware label smoothing" is not yet rigorously explained.

a) The paper claims that the smoothing matrix is ​​adaptively generated at the batch level, meaning that soft labels depend on the text similarity between samples within the same mini-batch. This may miss cases of "highly semantically similar samples outside the batch."

b) Does the smoothing mechanism truly model semantic overlap? Is the so-called "text-aware" smoothing truly "overlap-aware" or merely a general "similarity-aware" approach? If it's the latter, then this method is likely closer to the conventional regularization technique of "text similarity-based label smoothing" than a specific modeling of the challenge of medical semantic redundancy.

4. In text-aware label smoothing, it appears to be a direct application of similarity-weighted soft objectives to reduce the negative impact of over-penalizing semantic similarity. This is not conceptually novel. Furthermore, this smoothing is described as "more in line with clinical semantic similarity," but without any relevant analysis of clinical annotations, it's simply over-packaging a training stability technique into a clinical semantic alignment model.

**Questions:**

1. The logical relationship between HIP-InfoNCE and label smoothing seems confusing. In the Fig.2, the authors directly label "Smoothed Labels" next to "Loss: HIP-InfoNCE," easily leading one to mistakenly believe that HIP-InfoNCE itself includes a label smoothing mechanism. They fail to distinguish between the two stages of "similarity construction in HIP-InfoNCE" and "label smoothing in Text-aware Label Smoothing," which are not actually part of the bidirectional arrow relationship shown in the figure.
2. Contribution 1 is described as "analyzing the challenges of cross-modal retrieval in medicine."  Please explain why this is an independent contribution rather than a motivational stepping stone for subsequent modules.  It currently appears to be primarily a phenomenological description;  does the analysis itself contain any new quantitative findings?
3. Compared to existing sentence-level alignment, local alignment, and multi-view contrastive learning that treats multiple positive samples as the same semantic object after different image cropping, what is the core innovation of HIP-InfoNCE? From an implementation perspective, randomly masking reported fragments and aligning them with the whole image can be considered a form of textual data augmentation, rather than a new objective function design.
4. In label smoothing, does the smoothing mechanism mathematically reflect this semantic overlap, or does it merely apply a weaker penalty to semantically similar samples during training? Does the model treat these samples as "secondary positive samples" to bring them closer, or does it simply "not treat them as strong negative samples"?
5. The smoothing matrix is ​ adaptively generated within the mini-batch in this paper. This means that soft label assignment depends on the sample similarity visible within the batch. If two cases are globally semantically highly similar but never appear in the same batch, can the model still correctly learn their proximity relationship? Will this introduce training bias or instability? Have the authors considered alternatives based on memory queues or global caching?

---

### Official Review · Reviewer_rofx · 2025-10-31

**Soundness:** 3
**Presentation:** 3
**Contribution:** 2
**Rating:** 4
**Confidence:** 4

**Summary:**

This paper presents a study on a medical CLIP model, primarily identifying two challenges faced by current medical CLIP frameworks when trained on the MIMIC-CXR dataset: (1) excessively long medical report texts and (2) high similarity among reports. The authors propose corresponding solutions for both challenges and validate the effectiveness of their methods through experimental evaluation.

**Strengths:**

1. The analysis of the current challenges in medical CLIP models is compelling and well-supported by Fig. 1.
2. The experiments include a rich set of baselines and conduct comprehensive comparisons across both in-domain and out-of-domain settings.
3. The writing is clear and highly readable.
4. Numerous visualization results are provided, demonstrating the model’s superiority from an interpretability perspective.

**Weaknesses:**

1. I am confused by the analysis regarding "Holistic image versus multi-faceted report." When textual reports contain broader and more fine-grained information, shouldn’t this benefit the visual encoder by enabling it to learn richer, multi-dimensional representations? The observed performance gain seems attributable mainly to random masking as a form of data augmentation. In my view, text masking might be equal to image masking, which has already been explored in prior work [1], and the proposed HIP-InfoNCE loss appears to be an extension of such methods.
2. Another key contribution, Text-aware Label Smoothing, is presented as an extension of MedCLIP. The Related Work section also discusses MedCLIP in detail. Given its relevance, it should not be that MedCLIP is not included as a baseline in the experiments.
3. While the paper investigates the impact of different label smoothing strategies, it omits comparisons with MedCLIP’s approach. Additionally, although different BERT variants are evaluated for computing BERTScore, it would be valuable to also include other text encoders from medical CLIP models (e.g., BiomedCLIP).
4. The visualizations primarily rely on attention-based case studies. However, a global t-SNE analysis of embeddings on the MIMIC-CXR dataset would be more informative and better reflect the overall model behavior.
5. The experimental evaluation focuses solely on cross-modal retrieval, albeit across multiple in-domain and out-of-domain datasets. By convention, linear probing, zero-shot, and few-shot classification experiments are essential and should be included.

**Questions:**

1. Please clarify the concern raised in Weakness #1—specifically, why multi-faceted reports are detrimental—and further articulate the novelty of the HIP-InfoNCE loss beyond existing masked data augmentation techniques.
2. Explain why MedCLIP’s performance was not reported in the experiments.
3. Add t-SNE visualizations of the learned embeddings on MIMIC-CXR.
4. If time permits, please include results for linear probing, zero-shot, and few-shot classification tasks.
5. Since all analyses are based exclusively on MIMIC-CXR, it would be insightful to examine whether the identified challenges, high inter-report similarity and excessive text length, also manifest in other medical multimodal datasets, such as Quilt-1M [2] or PMC-15M [3].

If most of these concerns can be adequately addressed and the suggested experiments added, I would consider raising my score to 6. However, I am unlikely to rate it higher, as I find the overall novelty of the work somewhat limited.

[1] Chen, C., Zhong, A., Wu, D., Luo, J., & Li, Q. (2023, October). Contrastive masked image-text modeling for medical visual representation learning. In *International conference on medical image computing and computer-assisted intervention* (pp. 493-503). Cham: Springer Nature Switzerland.

[2] Ikezogwo, W., Seyfioglu, S., Ghezloo, F., Geva, D., Sheikh Mohammed, F., Anand, P. K., ... & Shapiro, L. (2023). Quilt-1m: One million image-text pairs for histopathology. *Advances in neural information processing systems*, *36*, 37995-38017.

[3] Zhang, S., Xu, Y., Usuyama, N., Xu, H., Bagga, J., Tinn, R., ... & Poon, H. (2023). Biomedclip: a multimodal biomedical foundation model pretrained from fifteen million scientific image-text pairs. *arXiv preprint arXiv:2303.00915*.

---

### Official Review · Reviewer_L7Ue · 2025-11-01

**Soundness:** 2
**Presentation:** 1
**Contribution:** 1
**Rating:** 2
**Confidence:** 5

**Summary:**

The paper presents a framework for medical image-report retrieval that addresses domain-specific limitations of existing CLIP-based methods. Its two contributions are HIP-InfoNCE loss and text-aware label smoothing, which benefit the model to better capture complex image-report relationships. Extensive experiments demonstrate its superior performance.

**Strengths:**

1. The problem of medical vision–language pretraining is important and clinically relevant.

2. The proposed approach achieves strong performance across multiple retrieval tasks.

**Weaknesses:**

1. The paper’s two main components—HIP-InfoNCE and text-aware label smoothing—are not novel. Very similar ideas exist in prior work: GLoRIA samples text spans from full reports for contrastive learning [1], and MedCLIP uses soft labels within a contrastive objective [2].

2. The study focuses primarily on retrieval. To more fully assess representation quality and practical utility, consider adding standard evaluations such as zero-shot classification (i.e., unseen-class recognition), linear probing on labeled datasets, and—where relevant—segmentation. Clear protocols and dataset details would strengthen the evidence.

3. The “masked views” in Eq. (1) are not described in sufficient detail. Please clarify how tokens are masked (random replacement vs. dropout), whether mask tokens are learnable, whether a single or multiple mask tokens are used, the mask rate/schedule, and whether masking is applied symmetrically across modalities. Ablations on these choices would help interpret results.

[1]. Shih-Cheng Huang, Liyue Shen, Matthew P Lungren, and Serena Yeung. GLoRIA: A multimodal
global-local representation learning framework for label-efficient medical image recognition. In
ICCV, 2021.

[2]. Zifeng Wang, Zhenbang Wu, Dinesh Agarwal, and Jimeng Sun. MedCLIP: Contrastive learning
from unpaired medical images and text. In EMNLP, 2022b.

**Questions:**

1. In Line 189, BLEU-4 emphasizes lexical overlap and may not reflect semantic alignment. Consider reporting metrics more aligned with meaning (e.g., BERTScore, Sentence-BERT cosine, CLIPScore) and radiology-specific measures (e.g., RadGraph-F1, CheXbert-F1) alongside BLEU-4, or justify its use for this task.

2. In Table 5, the proposed approach appears to generalize poorly across settings. Please analyze potential causes (data shift, prompt mismatch, overfitting) and, if possible, add cross-dataset evaluations or calibration results.

3. Figure 2 looks preliminary and is hard to follow. A clearer, professional schematic showing data flow, module boundaries, loss terms (including HIP-InfoNCE and label smoothing), and consistent notation would improve readability.

---

### Note · Authors · 2025-11-13

I have read and agree with the venue's withdrawal policy on behalf of myself and my co-authors.